# Learning from Integral Losses in Physics Informed Neural Networks

## Abstract

This work proposes a solution for the problem of training physics-informed networks under partial integro-differential equations. These equations require an infinite or a large number of neural evaluations to construct a single residual for training. As a result, accurate evaluation may be impractical, and we show that naive approximations at replacing these integrals with unbiased estimates lead to biased loss functions and solutions. To overcome this bias, we investigate three types of potential solutions: the deterministic sampling approach, the double-sampling trick, and the delayed target method. We consider three classes of PDEs for benchmarking; one defining Poisson problems with singular charges and weak solutions of up to 10 dimensions, another involving weak solutions on electromagnetic fields and a Maxwell equation, and a third one defining a Smoluchowski coagulation problem. Our numerical results confirm the existence of the aforementioned bias in practice, and also show that our proposed delayed target approach can lead to accurate solutions with comparable quality to ones estimated with a large number of samples. Our implementation is open-source and available at https://anonymous.4open.science/r/btspinn.

## 1 Introduction

Physics Informed Neural Networks (PINNs) (Raissi et al., 2019) can be described as solvers of a particular Partial Differential Equation (PDE). Typically, these problems consist of three defining elements. A sampling procedure selects a number of points for learning. Automatic differentiation is then used to evaluate the PDE at these points and define a residual. Finally, a loss function, such as the Mean Squared Error (MSE), is applied to these residuals, and the network learns the true solution by minimizing this loss through back-propagation and stochastic approximation. These elements form the basis of many methods capable of learning high-dimensional parameters. A wealth of existing work demonstrated the utility of this approach to solving a wide array of applications and PDE forms (Li et al., 2020; Shukla et al., 2021; Li et al., 2019).

One particular problem in this area, is the prevalent assumption around our ability to accurately evaluate the PDE residuals for learning. In particular, partial integro-differential forms include integrals or large summations within them. These forms appear in a broad range of scientific applications including quantum physics (Laskin, 2000), aerosol modeling (Wang et al., 2022a), and ecology (Humphries et al., 2010). In such instances, an accurate evaluation of the PDE elements, even at a single point, can become impractical. Naive approximations, such as replacing integrals with unbiased estimates, can result in biased solutions, as we will show later. This work is dedicated to the problem of learning PINNs with loss functions containing a parametrized integral or summation.

One natural approach for learning PINNs with integral forms would be to use techniques such as importance sampling or Quasi Monte Carlo (QMC) to estimate the integrals more accurately than a standard i.i.d. sampling approach. This follows the classical theory and such approaches have been investigated thoroughly in prior work (Caflisch, 1998; Evans and Swartz, 1995).

In this paper, we consider an alternative approach, which we will show can be more effective than reducing the variance of the integral estimation. The methods we investigate are based around the idea of reducing the bias and the variance in the parameter gradients, so that we can train effectively even if our loss functions are not accurately estimated. We consider three potential approaches to do this; the deterministic sampling approach, the double-sampling trick, and the delayed target method.

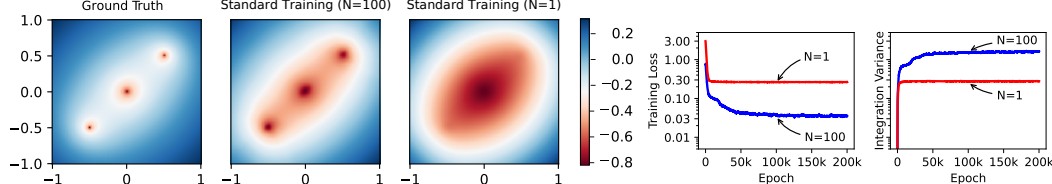

Figure 1: Training with the MSE loss under different sample sizes per surface ($N$). The heatmaps show the analytical solution (left), the low-variance training with $N = 100$ (middle), and the high-variance training with $N = 1$ (right). The smaller the $N$, the more biased the training objective becomes towards finding smoother solutions. The right panel shows the training curves; the training loss and the integration variance represent $\hat{\mathcal{L}}_\theta(x)$ and $\mathbb{V}_{P(x'|x)}[g_\theta(x')]$ in Equation 15, respectively. For $N = 1$, the training loss seems to be floored at the same value as the integration variance (i.e., approximately 0.3). However, with $N = 100$, the model produces better solutions, lower training losses, and higher integration variances.

As we will see, the delayed target approach, which is based upon ideas from learning temporal differences (Sutton, 1984; Mnih et al., 2015; Fujimoto et al., 2018), gives the best results, performing comparable or slightly better than accurate integral estimators (i.e., with $N = 100$ samples) using just a single sample ($N = 1$). Combining importance sampling and QMC methods with our techniques is a promising direction that we leave for future work.

The main contributions of this work are: (1) we formulate the integral learning problem under a general framework and show the biased nature of standard approximated loss functions; (2) we present three techniques to solve such problems, namely the deterministic sampling approach, the double-sampling trick, and the delayed target method; (3) we detail an effective way of implementation for the delayed target method compared to a naive one; and (4) we compare the efficacy of the potential solutions using numerical examples on Poisson problems with singular charges and up to 10 dimensions, a Maxwell problem with magnetic fields, and a Smoluchowski coagulation problem.

## 2 PROBLEM FORMULATION

Consider a typical partial integro-differential equation

$$f_\theta(x) := \mathbb{E}_{P(x'|x)}[g_\theta(x')] + y. \tag{1}$$

The $f_\theta(x)$ and $g_\theta(x')$ are parametrized, and $y$ includes all the non-parametrized terms in the PDE. The right side of the equation serves as the target value for $f_\theta(x)$. For notation simplicity, we assume a fixed value for $x$ in the remainder of the manuscript. However, all analyses are applicable to randomized $x$ without any loss of generality. Equation 1 is a general, yet concise, form for expressing partial integro-differential equations. To motivate, we will express three examples.

**Example 1** The Poisson problem is to solve the system $\nabla^2 U = \rho$ for $U$ given a charge function $\rho$. This is equivalent to finding a solution for the system

$$E = \nabla U, \qquad \nabla \cdot E = \rho. \tag{2}$$

A weak solution can be obtained through enforcing the divergence theorem over many volumes:

$$\int_{\partial\Omega} E \cdot \hat{n} \quad \mathrm{d}S = \iint_\Omega \nabla \cdot E \quad \mathrm{d}V, \tag{3}$$

where $\hat{n}$ is the normal vector perpendicular to $\mathrm{d}S$. The weak solutions can be preferable over the strong ones when dealing with singular or sparse $\rho$ charges.

To solve this system, we parametrize $E(x)$ as the gradient of a neural network predicting the $U$ potentials. To convert this into the form of Equation 1, we replace the left integral in Equation 3 with an arbitrarily large Riemann sum and write

$$\int_{\partial\Omega} E \cdot \hat{n} \quad \mathrm{d}S = \frac{A}{M} \sum_{i=1}^{M} E_\theta(x_i) \cdot \hat{n}_i, \tag{4}$$

---

**Algorithm 1** Regularized delayed target method for training integro-differential PINNs

---

**Require:** The initial parameter values $\theta_0$, learning rate $\eta$, Polyak averaging rate $\gamma$, target sample size $N$, and the target regularization weight $\lambda$.

1: Initialize the main and target parameters: $\theta, \theta_{\text{Target}} \leftarrow \theta_0$.
2: **for** $k = 1, 2, \ldots$ **do**
3:     Compute the $f_\theta(x)$ term using the main parameters.
4:     Sample the $\{x_i'\}_{i=1}^N$ set from $P(\{x_i'\}|x)$.
5:     Compute the target $\frac{1}{N}\sum_{i=1}^N g_{\theta^{\text{Target}}}(x_i') + y$ using the target parameters $\theta_{\text{Target}}$.
6:     Construct the main loss: $\hat{\mathcal{L}}_\theta^{\text{DT}} = \left(f_\theta(x) - \frac{1}{N}\sum_{i=1}^N g_{\theta^{\text{Target}}}(x_i') - y\right)^2$.
7:     Construct the target regularization loss: $\hat{\mathcal{L}}_\theta^{\text{R}} = (f_\theta(x) - f_{\theta^{\text{Target}}}(x))^2$.
8:     Compute the total loss $\hat{\mathcal{L}}_\theta^{\text{DTR}} = \hat{\mathcal{L}}_\theta^{\text{DT}}(x) + \lambda\hat{\mathcal{L}}_\theta^{\text{R}}$.
9:     Perform a gradient descent step on $\theta$: $\theta \leftarrow \theta - \eta\nabla_\theta\hat{\mathcal{L}}_\theta^{\text{DTR}}$.
10:    Update the target parameters using Polyak averaging: $\theta^{\text{Target}} \leftarrow \gamma\theta^{\text{Target}} + (1 - \gamma)\theta$.
11: **end for**

---

where $A = \int_{\partial\Omega} 1 \, \mathrm{d}S$ is the surface area and the $x_i$ samples are uniform on the surface. To convert this system into the form of Equation 1, we then define

$$x := x_1, \qquad f_\theta(x) := \frac{A}{M}E_\theta(x) \cdot \hat{n}_1, \qquad x' \sim \text{Unif}(\{x_i\}_{i=2}^M),$$

$$g_\theta(x_i) := -\frac{A(M-1)}{M}E_\theta(x_i) \cdot \hat{n}_i, \qquad y := \iint_\Omega \rho \mathrm{d}V. \tag{5}$$

**Example 2** In static electromagnetic conditions, one of the Maxwell Equations, the Ampere circuital law, is to solve the system $\nabla \times A = B$, $\nabla \times B = J$ for $A$ given the current density $J$ in the 3D space (we assumed a unit physical coefficient for simplicity). Here, $B$ represents the magnetic field and $A$ denotes the magnetic potential vector. A weak solution for this system can be obtained by enforcing the Stokes theorem over many volumes:

$$\int_{\partial\Omega} \nabla \times A \cdot \mathrm{d}\mathbf{l} = \iint_\Omega J \cdot \mathrm{d}S, \tag{6}$$

where $\mathrm{d}S$ is an infinitesimal surface normal vector. Just like the Poisson problem, the weak solutions can be preferable over the strong ones when dealing with singular inputs, and this equation can be converted into the form of Equation 1 similarly.

**Example 3** The Smoluchowski coagulation equation simulates the evolution of particles into larger ones, and is described as

$$\frac{\partial\rho(x,t)}{\partial t} = \int_0^x K(x-x',x')\rho(x-x',t)\rho(x',t)\mathrm{d}x' - \int_0^\infty K(x,x')\rho(x,t)\rho(x',t)\mathrm{d}x', \tag{7}$$

where $K(x,x')$ is the coagulation kernel between two particles of size $x$ and $x'$. The particle sizes $x$ and $x'$ can be generalized into vectors, inducing a higher-dimensional PDE to solve. To solve this problem, we parametrize $\rho(x,t)$ as the output of a neural network with parameters $\theta$ and write

$$f_\theta(x) := \frac{\partial\rho_\theta(x,t)}{\partial t}, \qquad g_\theta^{(1)}(x') := A_1 K(x-x',x')\rho_\theta(x-x',t)\rho_\theta(x',t),$$

$$g_\theta^{(2)}(x') := A_2 K(x,x')\rho_\theta(x,t)\rho_\theta(x',t). \tag{8}$$

The $x'$ values in both $g_\theta^{(1)}$ and $g_\theta^{(2)}$ are sampled from their respective uniform distributions, and $A_1$ and $A_2$ are used to normalize the uniform integrals into expectations. Finally, $y = 0$ and we can define $g_\theta(x')$ in a way such that

$$\mathbb{E}_{x'}[g_\theta(x')] := \mathbb{E}_{x'}[g_\theta^{(1)}(x')] + \mathbb{E}_{x'}[g_\theta^{(2)}(x')]. \tag{9}$$

The standard way to solve systems such as Examples 1, 2, and 3 with PINNs, is to minimize the following mean squared error (MSE) loss (Raissi et al., 2019; Jagtap et al., 2020):

$$\mathcal{L}_\theta(x) := \left(f_\theta(x) - \mathbb{E}_{P(x'|x)}[g_\theta(x')] - y\right)^2. \tag{10}$$

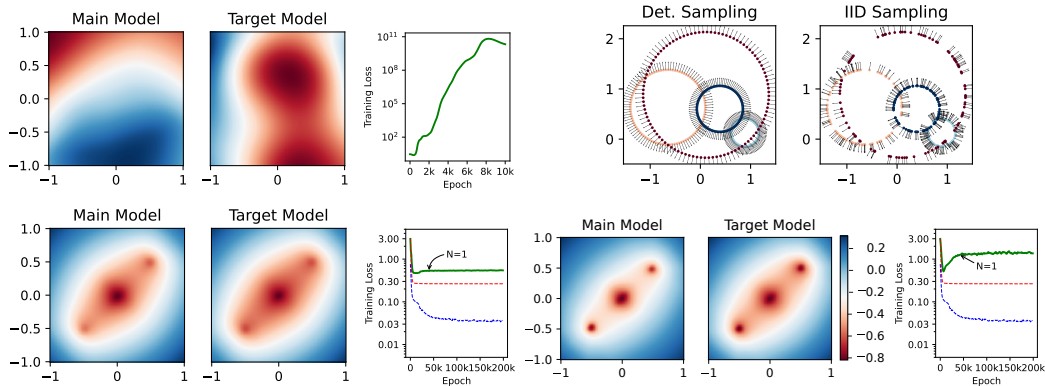

Figure 2: Training the same problem as in Figure 1 with delayed targets and $N = 1$. *The top left panel* shows a diverged training with $M = 100$ in Equation 24. *The lower left panel* corresponds to $M = 10$, which has a converging training curve even though it produces an overly smooth solution. In *the lower right panel*, we set $\lambda = 1$ which allowed setting the simulated $M$ as 1000 while maintaining a stable training loss. In each panel, the left and right heatmaps show the main and the target model predictions, respectively, and the right plots show the training curves. The green curves show the training loss for the delayed target method, and the standard training curves with $N = 1$ and 100 are also shown using dotted red and blue lines for comparison, respectively. *The top right panel* shows an example of deterministic vs. i.i.d. sampling of the surface points $\{x_i'\}_{i=1}^N$ in the Poisson problem. For each sampled sphere, the surface points and their normal vectors are shown with $N = 100$ samples. With deterministic sampling, the points are evenly spaced to cover the sampling domain.

Since computing exact integrals may be impractical, one may contemplate replacing the expectation in Equation 10 with an unbiased estimate, as implemented in NVIDIA's Modulus package (NVIDIA, 2022). This prompts the following approximate objective:

$$\hat{\mathcal{L}}_\theta(x) := \mathbb{E}_{\{x_i'\}_{i=1}^N}\left[\left(f_\theta(x) - \frac{1}{N}\sum_{i=1}^N g_\theta(x_i') - y\right)^2\right]. \tag{11}$$

We therefore analyze the approximation error by adding and subtracting $\mathbb{E}_{x''}[g_\theta(x'')]$:

$$\hat{\mathcal{L}}_\theta(x) = \mathbb{E}_{\{x_i'\}}\left[\left(\left(f_\theta(x) - \mathbb{E}_{x''}[g_\theta(x'')] - y\right) + \left(\mathbb{E}_{x''}[g_\theta(x'')] - \frac{1}{N}\sum_{i=1}^N g_\theta(x_i')\right)\right)^2\right]. \tag{12}$$

By decomposing the squared sum, we get

$$\hat{\mathcal{L}}_\theta(x) = \mathbb{E}_{\{x_i'\}}\left[\left(\left(f_\theta(x) - \mathbb{E}_{x''}[g_\theta(x'')] - y\right)^2\right] + \mathbb{E}_{\{x_i'\}}\left[\left(\mathbb{E}_{x''}[g_\theta(x'')] - \frac{1}{N}\sum_{i=1}^N g_\theta(x_i')\right)^2\right]$$

$$+ 2\mathbb{E}_{\{x_i'\}}\left[\left(f_\theta(x) - \mathbb{E}_{x''}[g_\theta(x'')] - y\right)\left(\mathbb{E}_{x''}[g_\theta(x'')] - \frac{1}{N}\sum_{i=1}^N g_\theta(x_i')\right)\right]. \tag{13}$$

Since $\mathbb{E}_{x''}[g_\theta(x'')] = \mathbb{E}_{\{x_i'\}}[\frac{1}{N}\sum_{i=1}^N g_\theta(x_i')]$, the last term in Equation 13 is zero, and we have

$$\hat{\mathcal{L}}_\theta(x) = \mathcal{L}_\theta(x) + \mathbb{V}_{P(\{x_i'\}|x)}[\frac{1}{N}\sum_{i=1}^N g_\theta(x_i')], \tag{14}$$

where $\mathbb{V}_P[x] := \mathbb{E}_P[x^2] - \mathbb{E}_P[x]^2$. If $x_i'$ are sampled in an i.i.d. manner, Equation 14 simplifies further to

$$\hat{\mathcal{L}}_\theta(x) = \mathcal{L}_\theta(x) + \frac{1}{N}\mathbb{V}_{P(x'|x)}[g_\theta(x')]. \tag{15}$$

The induced excess variance in Equation 15 can bias the optimal solution. As a result, optimizing the approximated loss will prefer smoother solutions over all $\{x_i'\}_{i=1}^N$ samples. It is worth noting that this bias is mostly harmful due to its parametrized nature; the only link through which this bias can offset the optimal solution is its dependency on $\theta$. This is in contrast to any non-parametrized stochasticity in the $y$ term of Equation 10. Non-parameterized terms cannot offset the optimal solutions, since stochastic gradient descent methods are indifferent to them.

Figure 3: The results of the deterministic and double sampling techniques on the Poisson problem. The left plots demonstrate the solutions with $N = 1$, while the right plots show the solutions with $N = 100$. The training curves represent the mean squared error to the analytical solution vs. the training epochs. With $N = 1$, the double sampling trick exhibits divergence in training, and the deterministic sampling process yields overly smooth functions similar to the standard solution in Figure 1. However, with $N = 100$, both the deterministic and double-sampling approaches exhibit improvements. According to the training curves, the delayed target method with $N = 1$ yields the best solutions in this problem.

## 3    POTENTIAL SOLUTIONS

Based on Equation 15, the induced bias in the solution has a direct relationship with the stochasticity of the conditional distribution $P(x'|x)$. If we were to sample the $(x, x')$ pairs deterministically, the excess variance in Equation 15 would disappear. However, this condition is satisfied only by modifying the problem conditions. Next, we introduce three potential solutions to this problem: the *deterministic sampling strategy*, the *double-sampling trick*, and the *delayed target method* which is based upon the method of learning from temporal differences (Sutton, 1984).

**The deterministic sampling strategy**: One approach to eliminate the excess variance term in Equation 14, is to sample the $\{x'_i\}_{i=1}^N$ set in a way that $P(\{x'_i\}|x)$ would be a point mass distribution at a fixed set $\mathbb{A}_x$. This way, $P(\{x'_i\}|x)$ yields a zero excess variance: $\mathbb{V}[\frac{1}{N}\sum_{i=1}^N g_\theta(\mathbb{A}_x^{(i)})] = 0$. This induces the following deterministic loss.

$$\hat{\mathcal{L}}_\theta^{\text{DET}}(x) := \left(f_\theta(x) - \frac{1}{N}\sum_{i=1}^N g_\theta(\mathbb{A}_x^{(i)}) - y\right)^2. \tag{16}$$

Although this approach removes the excess variance term in Equation 14 thanks to its deterministic nature, it biases the optimization loss by re-defining it: $\mathcal{L}_\theta(x) \neq \hat{\mathcal{L}}_\theta^{\text{DET}}(x)$. The choice of the $\mathbb{A}_x$ set can influence the extent of this discrepancy. One reasonable choice is to evenly space the $N$ samples to cover the entire sampling domain as uniformly as possible. For a demonstration, Figure 2 shows a number of example sets used for applying the divergence theorem to the Poisson problem. Of course, this sampling strategy can be impractical in high-dimensional spaces, which could be partially ameliorated by the use of QMC methods, as the number of samples needed to cover the entire sampling domain grows exponentially with the sampling space dimension.

**The double-sampling trick**: If we have two independent $x'$ samples, namely $x'_1$ and $x'_2$, we can replace the objective in Equation 11 with

$$\hat{\mathcal{L}}_\theta^{\text{DBL}}(x) = \mathbb{E}_{x'_1, x'_2 \sim P(x'|x)}\left[\left(f_\theta(x) - g_\theta(x'_1) - y\right) \cdot \left(f_\theta(x) - g_\theta(x'_2) - y\right)\right]. \tag{17}$$

It is straightforward to show that $\hat{\mathcal{L}}_\theta^{\text{DBL}}(x) = \mathcal{L}_\theta(x)$; the uncorrelation between $g_\theta(x'_1)$ and $g_\theta(x'_2)$ will remove the induced bias on average. However, this approach requires access to two i.i.d. samples, which may not be plausible in many sampling schemes. In particular, Monte-Carlo samplings used in reinforcement learning do not usually afford the learning method with the freedom to choose multiple next samples or the ablity to reset to a previous state. Besides reinforcement learning, offline learning using a given collection of samples may make this approach impractical.

**The delayed target method**: This approach replaces the objective in Equation 11 with

$$\mathcal{L}_\theta^{\text{DT}}(x) = \mathbb{E}_{P(x'|x)}\left[\left(f_\theta(x) - g_{\theta^*}(x') - y\right)^2\right], \tag{18}$$

where we have defined $\theta^* := \arg\min_{\tilde{\theta}} \mathcal{L}_{\tilde{\theta}}(x)$. Assuming a complete function approximation set $\Theta$ (where $\theta \in \Theta$), we know that $\theta^*$ satisfies Equation 1 at all points. Therefore, we have

$$\nabla_\theta \mathcal{L}_\theta(x)\big|_{\theta=\theta^*} = \nabla_\theta \mathcal{L}_\theta^{\text{DT}}(x)\big|_{\theta=\theta^*} = 0. \tag{19}$$

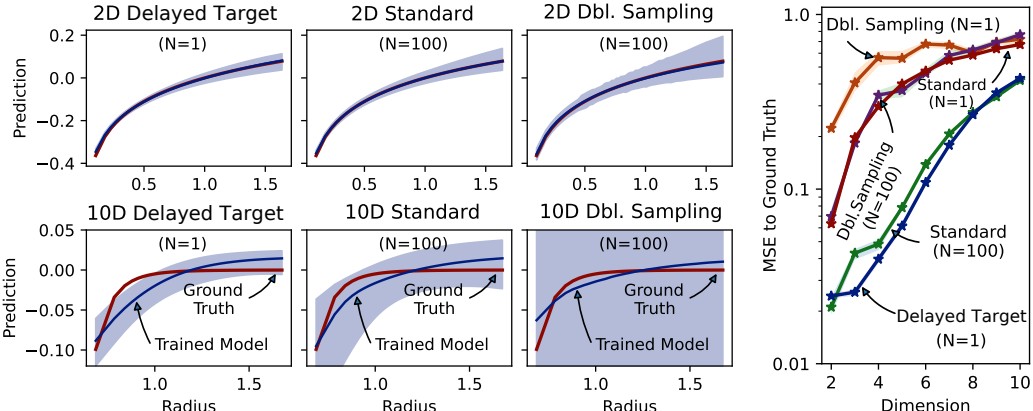

Figure 4: The solution and performance curves in higher-dimensional Poisson problems. *The left panel* shows the solution curves for the delayed target ($N = 1$), the standard ($N = 100$), and the double-sampling ($N = 100$) methods. The top and the bottom rows show 2- and 10-dimensional problems, respectively. In these problems, a single charge is located at the origin, so that the analytical solution is a function of the evaluation point radii $\|x\|$. The horizontal axis shows the evaluation point radii, and covers 99% of points within the training volumes. *The right chart* is a performance curve against the problem dimension (lower is better). The normalized MSE values were shown to be comparable. These results suggest that (1) higher dimensions make the problem challenging, and (2) delayed targeting with $N = 1$ is comparable to standard trainings with $N = 100$.

Therefore, we can claim

$$\theta^* = \arg\min_{\theta} \mathbb{E}_x[\mathcal{L}_{\theta}^{\text{DT}}(x)] = \arg\min_{\theta} \mathbb{E}_x[\mathcal{L}_{\theta}(x)]. \tag{20}$$

In other words, optimizing Equation 18 should yield the same solution as optimizing the true objective $\mathcal{L}_{\theta}(x)$ in Equation 10. Of course, finding $\theta^*$ is as difficult as solving the original problem. The simplest heuristic replaces $\theta^*$ with a supposedly independent, yet identically valued, version of the latest $\theta$ named $\theta^{\text{Target}}$, hence the delayed, detached, and bootstrapped target naming conventions:

$$\hat{\mathcal{L}}_{\theta}^{\text{DT}}(x) = \mathbb{E}_{P(\{x_i'\}|x)}\left[\left(f_{\theta}(x) - \frac{1}{N}\sum_{i=1}^{N} g_{\theta^{\text{Target}}}(x_i') - y\right)^2\right]. \tag{21}$$

Our hope would be for this approximation to improve as well as $\theta$ over training. The only practical difference between implementing this approach and minimizing the loss in Equation 10 is to use an incomplete gradient for updating $\theta$ by detaching the $g(x')$ node from the computational graph in the automatic differentiation software. This naive implementation of the delayed target method can lead to divergence in optimization, as we will show in Section 4 with numerical examples (i.e., Figure 2). Here, we introduce two mitigation factors contributing to the stabilization of such a technique.

**Moving target stabilization**   One disadvantage of the aforementioned technique is that it does not define a global optimization objective; even the average target for $f_{\theta}(x)$ (i.e., $\mathbb{E}\left[g_{\theta^{\text{Target}}}(x')\right] + y$) changes throughout the training. Therefore, a naive implementation can risk training instability or even divergence thanks to the moving targets.

To alleviate the fast-moving targets issue, prior work suggested fixing the target network for many time-steps (Mnih et al., 2015). This causes the training trajectory to be divided into a number of episodes, where the target is locally constant and the training is therefore locally stable in each episode. Alternatively, this stabilization can be implemented continuously using Polyak averaging (Fujimoto et al., 2018); instead of fixing the target network for a window of $T$ steps, the target parameters $\theta^{\text{Target}}$ can be updated slowly with the rule

$$\theta^{\text{Target}} \leftarrow \gamma\theta^{\text{Target}} + (1-\gamma)\theta. \tag{22}$$

This exponential moving average defines a corresponding stability window of $T = O(1/(1-\gamma))$.

**Prior imposition for highly stochastic targets**   In certain instances, the target $\frac{1}{N}\sum_{i=1}^{N} g_{\theta^{\text{Target}}}(x_i') + y$ in Equation 18 can be excessively stochastic, leading to divergence in the training of the delayed

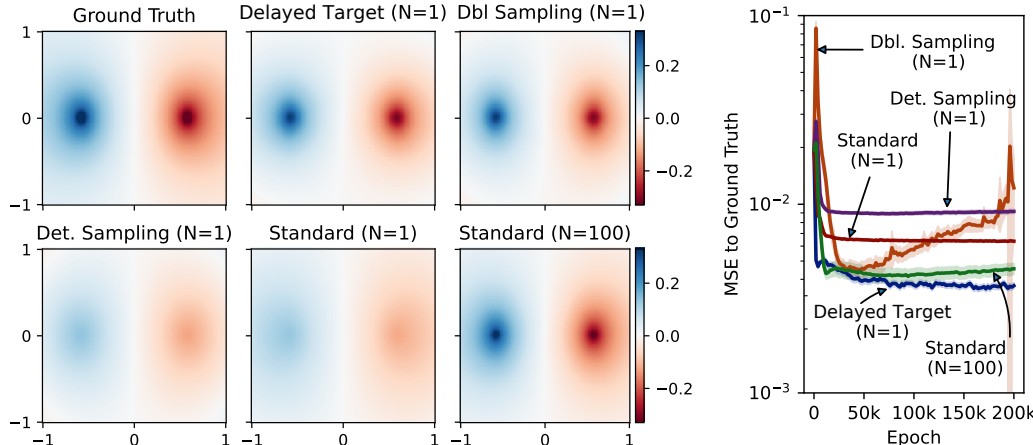

Figure 5: The solution heatmaps and the training curves for different methods to the Maxwell problem. *In the left panel*, we show a single component of the magnetic potentials ($A_z$) in a 2D slice of the training space with $z = 0$ for visual comparison. *In the right plot*, we show the training curves. The results suggest that (1) the standard and deterministic trainings with $N = 1$ produce overly smooth solutions, and (2) delayed targeting with $N = 1$ is comparable to standard trainings with $N = 100$.

target model. In particular, based on the settings defined in Equation 5 for the Poisson problem, we can write $g_\theta(x_i) = (M - 1) f_\theta(x_i)$. Therefore, we can analyze the target variance as

$$\mathbb{V}_{\{x_i'\}_{i=1}^N}\Big[\frac{1}{N}\sum_{i=1}^N g_{\theta^{\text{Target}}}(x_i') + y\Big] = \frac{(M-1)^2}{N}\mathbb{V}_{\{x_i'\}_{i=1}^N}[f_{\theta^{\text{Target}}}(x_i')] + \mathbb{V}[y]. \tag{23}$$

Ideally, $M \to \infty$ in order for Equation 4 to hold. Setting arbitrarily large $M$ will lead to unbounded target variances in Equation 23, which can slow down the convergence of the training or result in divergence. In particular, such unbounded variances can cause the main and the target models to drift away from each other leading to incorrect solutions.

One technique to prevent this drift, is to impose a Bayesian prior on the main and the target models. Therefore, to discourage this divergence phenomenon, we regularize the delayed target objective in Equation 24 and replace it with

$$\hat{\mathcal{L}}_\theta^{\text{DTR}} := \hat{\mathcal{L}}_\theta^{\text{DT}}(x) + \lambda \cdot (f_\theta(x) - f_{\theta^{\text{Target}}}(x))^2. \tag{24}$$

A formal description of the regularized delayed targeting process is given in Algorithm 1, which covers both the moving target stabilization and the Bayesian prior imposition.

## 4 EXPERIMENTS

We examine solving three problems. First, we solve a Poisson problem with singular charges using the divergence theorem as a proxy for learning. Figures 1, 2, and 3 define a 2D Poisson problem with three unit Dirac-delta charges at $[0, 0]$, $[-0.5, -0.5]$, and $[0.5, 0.5]$. We also study higher-dimensional Poisson problems with a unit charge at the origin in Figure 4. Our second example looks at finding the magnetic potentials and fields around a current circuit, which defines a singular current density profile $J$. Finally, to simulate particle evolution dynamics, we consider a Smoluchowski coagulation problem where particles evolve from an initial density. We designed the coagulation kernel $K$ to induce non-trivial solutions in our solution intervals. We employ a multi-layer perceptron as our deep neural network, using 64 hidden neural units in each layer, and the $\tanh$ activation function. We trained our networks using the Adam (Kingma and Ba, 2014) variant of the stochastic gradient descent algorithm under a learning rate of $0.001$. For a fair comparison, we afforded each method 1000 point evaluations for each epoch. Due to space limitations, a wealth of ablation studies with more datasets and other experimental details were left to the supplementary material.

**The Poisson problem with singular charges** To show the solution bias, we first train two models: one with $N = 100$ samples per sphere, and another one with only $N = 1$ sample per sphere. These

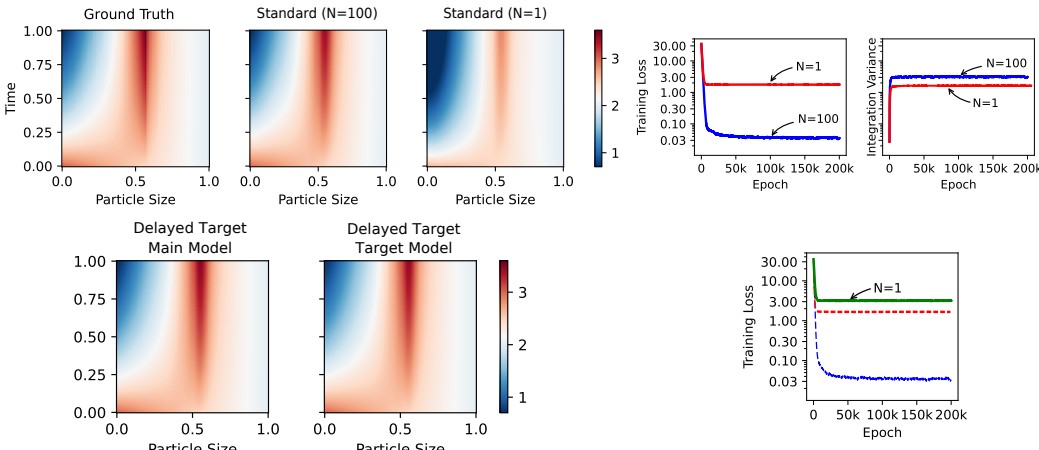

Figure 6: Training results on the Smoluchowski coagulation problem. The top left panel shows the ground truth solution, along with the standard $N = 100$ and $N = 1$ solution heatmaps minimizing the $\hat{\mathcal{L}}_\theta(x)$ in Equation 15. The training loss and the integration variance represent the $\hat{\mathcal{L}}_\theta(x)$ and $\mathbb{V}_{P(x'|x)}[g_\theta(x')]$ quantities in Equation 15. The top right figure shows the training curve for both of the standard trainings. The bottom left panel shows the delayed target solution heatmaps using $N = 1$ sample with its training curve next to it.

models represent a baseline for later comparisons. Based on Equation 15, the induced solution bias should be lower in the former scenario. Figure 1 shows the solution defined by these models along with the analytical solution and their respective training curves. The model trained with high estimation variance derives an overly smooth solution. We hypothesize that this is due to the excess variance in the loss. This hypothesis is confirmed by matching the training loss and the excess variance curves; the training loss of the model with $N = 1$ is lower bounded by its excess variance, although it successfully finds a solution with a smaller excess variance than the $N = 100$ model. An alternative capable of producing similar quality solutions with only $N = 1$ samples would be ideal.

To investigate the effect of highly stochastic targets on delayed target models, Figure 2 shows the training results with both $M = 100$ and $M = 10$. The former is unstable, while the latter is stable; this confirms the influence of $M$ in the convergence of the delayed target trainings. Furthermore, when this divergence happens, a clear drift between the main and the target models can be observed. Figure 2 shows that imposing the Bayesian prior of Equation 24 can lead to training convergence even with a larger $M = 1000$, which demonstrates the utility of our proposed solution.

We also investigated the performance of the deterministic and double-sampling techniques in this problem. Figure 3 shows these results when $N = 1$ and $N = 100$ samples are used for integral estimation. With $N = 1$, the training with the deterministic sampling approach is stable and yields similar results to those seen in Figure 1. The double-sampling trick, on the other hand, exhibits unstable trainings and sub-optimal solutions. We suspect that (a) the singular nature of the analytical solution, and (b) the stochasticity profile of the training loss function $\hat{\mathcal{L}}_\theta^{\text{DBL}}(x)$ in Equation 17 are two of the major factors contributing to this outcome. With $N = 100$, both the deterministic and double-sampling trainings yield stable training curves and better solutions. This suggests that both methods can still be considered as viable options for training integro-differential PINNs, conditioned on that the specified $N$ is large enough for these methods to train stably and well.

The regularized delayed target training with $N = 1$ sample is also shown in the training curves of Figure 3 for easier comparison. The delayed target method yields better performance than the deterministic or double-sampling in this problem. This may seemingly contradict the fact that the double-sampling method enjoys better theoretical guarantees than the delayed target method, since it optimizes a complete gradient. However, our results are consistent with recent findings in off-policy reinforcement learning; even in deterministic environments where the application of the double-sampling method can be facilitated with a single sample, incomplete gradient methods (e.g., TD-learning) may still be preferable over the full gradient methods (e.g., double-sampling) (Saleh and Jiang, 2019; Fujimoto et al., 2022; Yin et al., 2022; Chen et al., 2021). Intuitively, incomplete gradient methods detach parts of the gradient, depriving the optimizer from exercising full control over the decent direction and make it avoid over-fitting. In other words, incomplete gradient methods

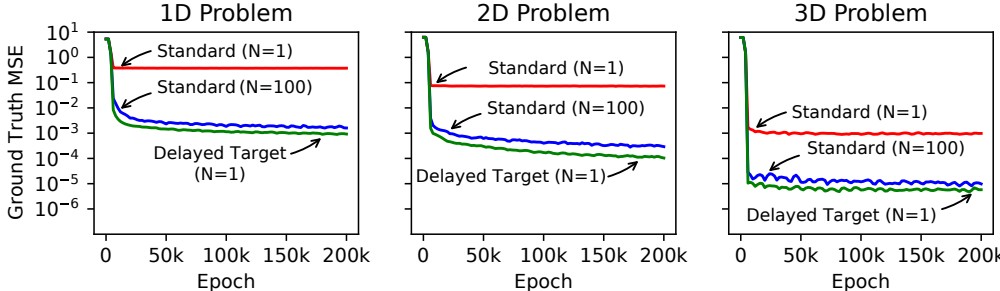

Figure 7: The solution mean squared error to the ground truth in the 1, 2, and 3-dimensional Smoluchowski coagulation problem. The vertical axis shows the solution error, and the horizontal axis shows the training epochs. The standard solutions were trained by the ordinary MSE loss $\mathcal{L}_\theta(x)$ in Equation 10 with $N = 1$ and $N = 100$ samples. The delayed target solution used $N = 1$ sample, yet produced slightly better results than the standard method with $N = 100$.

can be viewed as a middle ground between zero-order and first-order optimization, and may be preferable over both of them.

Figure 4 also studies the effect of problem dimensionality on our methods. The results confirm that the problem becomes significantly more difficult with higher dimensions. However, the delayed target models maintain comparable quality to standard trainings with large $N$.

**The Maxwell problem with a rectangular current circuit** Figure 5 shows the training results for the Maxwell problem. The results suggest that the standard and the deterministic trainings with small $N$ produce overly smooth solutions. The double-sampling method with small $N$ improves the solution quality at first, but has difficulty maintaining a stable improvement. However, delayed targeting with small $N$ seems to produce comparable solutions to the standard training with large $N$.

**The Smoluchowski coagulation problem** Figure 6 shows the training results for the Smoluchowski coagulation problem. Similar to the results in Figure 1, the standard training using $N = 1$ sample for computing the residual summations leads to biased and sub-optimal solution. However, the standard training with $N = 100$ samples suffers less from the effect of bias. The delayed target solution using only $N = 1$ sample produces comparable solution quality to the standard evaluation with $N = 100$ and is not bottlenecked by the integration variance. Figure 7 compares the solution quality for each of the standard and delayed target methods under different problem dimensions. The results suggest that the delayed target solution maintains its quality even in higher dimensional problems, where the excess variance issue leading to biased solutions may be more pronounced.

**Recommendations and limitations** From the above examples, we consistently see that the delayed target method shows superior performance over the other methods. However, this also has limitations, as this method is more temperamental than the standard trainings, and may require careful specification of hyper-parameters such as $\lambda$ and $\gamma$, as we showed in Figure 2.

## 5 CONCLUSION

In this work, we investigated the problem of learning PINNs in partial integro-differential equations. We presented a general framework for the problem of learning from integral losses, and theoretically showed that naive approximations of the parametrized integrals lead to biased loss functions due to the induced excess variance term in the optimization objective. We confirmed the existence of this issue in numerical simulations. Then, we studied three potential solutions to account for this issue, and we found the delayed target method to perform best in a wide class of problems. Our numerical results support the utility of this method on three classes of problems, (1) Poisson problems with singular charges and up to 10 dimensions, (2) an electromagnetic problem under a Maxwell equation, and (3) a Smoluchowski coagulation problem. The delayed target method. The limitations of our work include its narrow scope to learning PINNs; this work could have broader applications in other areas of machine learning. Also, future work should consider the applications of the delayed target method to more problem classes both in scientific and traditional machine learning areas. Developing adaptive processes for setting each method's hyper-parameters, and combining importance sampling or QMC techniques with our methods, are two other worthwhile future endeavors.

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

# A    SUPPLEMENTARY MATERIAL

## A.1    PROBABILISTIC AND MATHEMATICAL NOTATIONS.

We denote expectations with $\mathbb{E}_{P(z)}[h(z)] := \int_z h(z)P(z)\mathrm{d}z$, and variances with $\mathbb{V}_{P(z)}[h(z)] := \mathbb{E}_{P(z)}[h(z)^2] - \mathbb{E}_{P(z)}[h(z)]^2$ notations. Note that only the random variable in the subscript (i.e., $z$) is eliminated after the expectation. The set of samples $\{x'_1, \cdots x'_n\}$ is denoted with $\{x'_i\}_{i=1}^N$, and we abuse the notation by replacing $\{x'_i\}_{i=1}^N$ with $\{x'_i\}$ for brevity. Throughout the manuscript, $f_\theta(x)$ denotes the output of a neural network, parameterized by $\theta$, on the input $x$. The loss functions used for minimization are denoted with the $\mathcal{L}$ notation (e.g., $\mathcal{L}_\theta(x)$). $\nabla U := [\frac{\partial}{\partial x_1}U, \cdots, \frac{\partial}{\partial x_d}U]$ denotes the gradient of a scalar function $U$, $\nabla \cdot E := \frac{\partial E_1}{\partial x_1} + \cdots + \frac{\partial E_d}{\partial x_d}$ denotes the divergence of the vector field $E$, and $\nabla^2 U := \nabla \cdot \nabla U$ denotes the Laplacian of the function $U$. The $d$-dimensional Dirac-delta function is denoted with $\delta^d$, volumes are denoted with $\Omega$, and surfaces are denoted with $\partial\Omega$. The Gamma function is denoted with $\Gamma$, where $\Gamma(n) := (n-1)!$ for integer $n$. The uniform probability distribution over an area $A$ is denoted with $\mathrm{Unif}(A)$. These notations and operators are summarized in Tables 1 and 2.

| Notation | Description |
|---|---|
| $f_\theta(x)$ | The main neural output parameterized by $\theta$ |
| $g_\theta(x)$ | Secondary neural output parameterized by $\theta$ |
| $\mathcal{L}$ | Generic loss functions representation |
| $\mathcal{L}_\theta(x)$ | Loss $\mathcal{L}$ parametrized by $\theta$ evaluated at point $x$ |
| $\hat{\mathcal{L}}$ | Generic approximated loss representation |
| $N$ | Number of samples used for integral estimation |
| $\{x'_i\}$ | An abbreviation for the sample set $\{x'_i\}_{i=1}^N$ |
| $\delta^d$ | The $d$-dimensional Dirac-delta function |
| $\Omega$ | Volume representation |
| $\partial\Omega$ | Surface representation |
| $\Gamma$ | The Gamma function, where $\Gamma(n) := (n-1)!$ for integer $n$. |
| $\mathrm{Unif}(A)$ | The uniform probability distribution over an area or set $A$ |
| $\mathrm{Uniform}(A)$ | The same as $\mathrm{Unif}(A)$ |
| $U$ | The potential function in the Poisson Problem |
| $E$ | The gradient field in the Poisson Problem |
| $K$ | The Smoluchowski coagulation kernel used in Equation 7 |
| $\rho$ | Particle densities in the Smoluchowski equation |
| $A$ | The magnetic potentials in the Maxwell-Ampere problem |
| $B$ | The magnetic field in the Maxwell-Ampere problem |
| $J$ | The current density field in the Maxwell-Ampere problem |
| $I$ | The current flowing through a plane in the Maxwell-Ampere problem |
| $\gamma$ | The delayed target Polyak averaging factor in Algorithm 1 |
| $\lambda$ | The delayed target regularization weight defined in Equation 24 |

Table 1: The mathematical notations used throughout the paper.

| Notation | Definition | Description |
|---|---|---|
| $\mathbb{E}_{P(z)}[h(z)]$ | $\int_z h(z)P(z)\mathrm{d}z$ | Expectation of $h(z)$ over $z \sim P(\cdot)$ |
| $\mathbb{V}_{P(z)}[h(z)]$ | $\mathbb{E}_{P(z)}[h(z)^2] - \mathbb{E}_{P(z)}[h(z)]^2$ | Variance of $h(z)$ over $z \sim P(\cdot)$ |
| $\{x_i'\}_{i=1}^N$ | $\{x_1', x_2', \cdots x_n'\}$ | Generic sample set |
| $\nabla U$ | $[\frac{\partial}{\partial x_1}U, \cdots, \frac{\partial}{\partial x_d}U]^\intercal$ | Gradient of the potential $U$ |
| $\nabla \cdot E$ | $\frac{\partial E_1}{\partial x_1} + \cdots + \frac{\partial E_d}{\partial x_d}$ | Divergence of the field $E$ |
| $\nabla^2 U$ | $\nabla \cdot \nabla U$ | Laplacian of the potential $U$ |
| $\nabla \times A$ | $\begin{bmatrix} \frac{\partial A_3}{\partial x_2} - \frac{\partial A_2}{\partial x_3} \\ \frac{\partial A_1}{\partial x_3} - \frac{\partial A_3}{\partial x_1} \\ \frac{\partial A_2}{\partial x_1} - \frac{\partial A_1}{\partial x_2} \end{bmatrix}$ | Curl of the 3D field $A$ |

Table 2: The mathematical operators used throughout the paper.

## A.2 RELATED WORK

**Integro-differential PDEs in scientific applications** Integro-differential PDEs arise in many areas such as quantum physics (Laskin, 2000; 2002; Elgart and Schlein, 2007; Lieb and Yau, 1987), visco-elastic fluid dynamics (Constantin, 2005; Caffarelli et al., 2011; Caffarelli and Vasseur, 2010a;b), nuclear reaction physics (Bern et al., 1994), mathematical finance (Nolan, 1999; Ros Oton, 2014), ecology (Humphries et al., 2010; Cabré and Roquejoffre, 2013; Reynolds and Rhodes, 2009; Viswanathan et al., 1996), elasticity and material modeling (Toland, 1997; Lu, 2005), particle system evolutions (Chapman et al., 1996; Weinan, 1994; Giacomin and Lebowitz, 1997; Carrillo et al., 2011), aerosol modeling (Wang et al., 2022a), computed tomography (Wei et al., 2019), radiation transfer and wave propagation (Modest and Mazumder, 2021), grazing systems and epidemealogy (Lakshmikantham, 1995), and in the formulation of weak solutions with methods such as variational PINNs (Kharazmi et al., 2021). Weak solutions using the divergence or the curl theorems, or the Smoluchowski coagulation equation (Wang et al., 2022a) are a few representative forms we consider in this work.

**Physics-informed networks** The original PINN was introduced in Raissi et al. (2019). Later, variational PINNs were introduced in Kharazmi et al. (2019). Variational PINNs introduced the notion of weak solutions using test functions into the original PINNs. This was later followed by hp-VPINNs (Kharazmi et al., 2021). In fact, integral forms appear in both VPINNs and hp-VPINNs, and our methods could be used synergistically with these variational models to improve them. Conservative PINNs (or cPINNs for short) were also proposed and used to solve physical systems with conservation laws (Jagtap et al., 2020) and Mao et al. (2020) examined the application of PINNs to high-speed flows. Many other papers attempted to scale and solve the fundamental problems with PINNs, for example, using domain decomposition techniques (Shukla et al., 2021; Li et al., 2019), the causality views (Wang et al., 2022b), and neural operators (Li et al., 2020). Reducing the bias of the estimated training loss is a general topic in machine and reinforcement learning (Sutton, 1988; Ghaffari et al., 2022; Arazo et al., 2020).

**Bootstrapping neural networks** The delayed target strategies have been looked at in other contexts, such as reinforcement or semi-supervised learning. The TD-learning method is an early example of this family (Sutton, 1984), and it has been analyzed extensively in prior work (Dayan, 1992; Tsitsiklis and Van Roy, 1997; Baird, 1995; Li, 2008; Schoknecht and Merke, 2003). Time and time again, TD-learning has proven preferable over the ordinary MSE loss minimization (known as the Bellman residual minimization) (Saleh and Jiang, 2019; Fujimoto et al., 2022; Yin et al., 2022; Chen et al., 2021). The deep Q-networks proposed a practical adaptation of this methodology (Mnih et al., 2015), which has been complemented in the TD3 method (Fujimoto et al., 2018). The top-left panel of

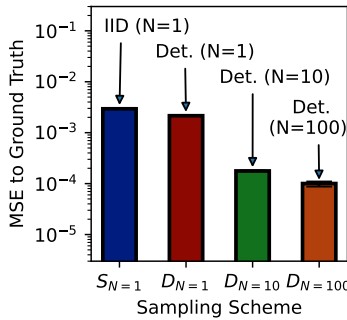 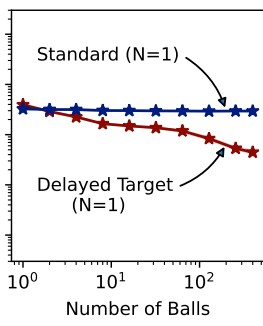 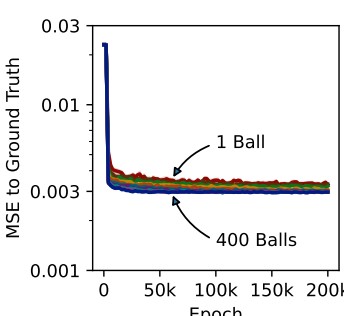

Figure 8: Ablation studies on the sampling hyper-parameters and settings. *In the left plot*, we compare the deterministic and i.i.d. sampling on the standard trainings with various $N$. *In the middle plot*, the horizontal axis shows the number of balls sampled in each epoch. Both the standard and the delayed target methods are shown in this plot with $N = 1$. *The right plot* shows the training curves for the standard method with $N = 1$ target samples. All experiments were conducted on the Poisson problem in Figure 1 of the main paper.

Figure 2 shows the closest algorithmic match to the TD3 value learning method in our setup, which produced a diverged training in our challenging and singular problems. Our work identified and mitigated this issue through prior imposition for highly stochastic targets.

Another example is the recent trend of semi-supervised learning, where teacher-student frameworks result in accuracy improvements of classification models by pseudo-labelling unlabelled examples for training (Hinton et al., 2015; Pham et al., 2021; Arazo et al., 2020; Lee et al., 2013). While a small number of recent theoretical insights exist on why semi-supervised learning does not produce trivially incorrect solutions (Tian et al., 2021), a wealth of theoretical literature analyzed the ability and shortcomings of TD-learning methods to solve such problems.

### A.3   THE ANALYTICAL SOLUTIONS

**The Poisson Problem**   Consider the $d$-dimensional space $\mathbb{R}^d$ and the following charge:

$$\rho_x = \delta^d(x). \tag{25}$$

For $d \neq 2$, the analytical solution to the system

$$\nabla \cdot E = \rho, \qquad \nabla U = E \tag{26}$$

can be derived as

$$U(x) = \frac{\Gamma(d/2)}{2 \cdot \pi^{d/2} \cdot (2-d)} \|x\|^{2-d}, \qquad E(x) = \frac{\Gamma(d/2)}{2 \cdot \pi^{d/2} \cdot \|x\|^d} x. \tag{27}$$

For $d = 2$, $E_x$ stays the same but we have $U(x) = \frac{1}{2\pi} \ln(\|x\|)$.

We want to solve this system using the divergence theorem:

$$\int_{\partial\Omega} E \cdot \hat{n} \quad \mathrm{d}S = \iint_\Omega \nabla \cdot E \quad \mathrm{d}V. \tag{28}$$

Keep in mind that the $d-1$-dimensional surface area of a $d$-dimensional sphere with radius $r$ is

$$A_d^r := \int_{\partial\Omega_r} 1 \, \mathrm{d}S = \frac{2 \cdot \pi^{d/2}}{\Gamma(d/2)} \cdot r^{d-1}. \tag{29}$$

**The Maxwell-Ampere Equation**   Consider the 3-dimensional space $\mathbb{R}^3$, and the following current along the z-axis:

$$J(x) = I \cdot \delta^2(x_1 = 0, x_2 = 0, x_3 \in [z_1, z_2]). \tag{30}$$

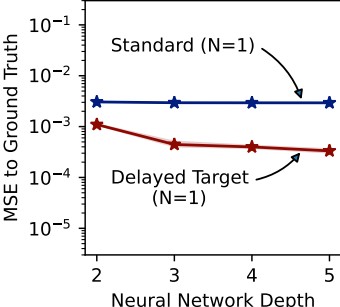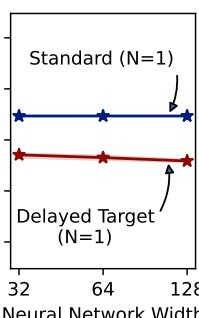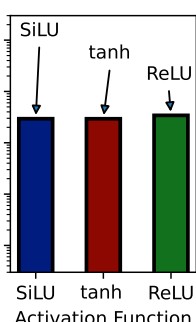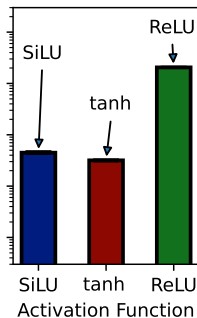

Figure 9: Ablating the function approximation class attributes on the 2D Poisson problem in Figure 1 of the main paper. A multi-layer perceptron was used in all of our experiments. *The left and right line plots* show the effect of the neural network's depth and width, respectively, on each of the standard and delayed target methods. *The left and right bar plots* demonstrate the effect of the neural activation function on the standard and the delayed target methods, respectively. These results indicate that the function approximation class can have a more substantial impact on the delayed target method than the standard trainings.

The analytical solution to the system

$$\nabla \times A = B, \qquad \nabla \times B = J \tag{31}$$

can be expressed as

$$A = \frac{-I}{4\pi} \cdot \log \left( \frac{(z_2 - x_3) + \sqrt{x_1^2 + x_2^2 + (z_2 - x_3)^2}}{(z_1 - x_3) + \sqrt{x_1^2 + x_2^2 + (z_1 - x_3)^2}} \right) \left[ 0, 0, 1 \right]^T, \tag{32}$$

and

$$B = \frac{-I}{4\pi \cdot \sqrt{x_1^2 + x_2^2}} \cdot \left( \frac{z_2 - x_3}{\sqrt{x_1^2 + x_2^2 + (z_2 - x_3)^2}} - \frac{z_1 - x_3}{\sqrt{x_1^2 + x_2^2 + (z_1 - x_3)^2}} \right) \cdot \begin{bmatrix} \frac{-x_2}{\sqrt{x_1^2 + x_2^2}} \\ \frac{x_1}{\sqrt{x_1^2 + x_2^2}} \\ 0 \end{bmatrix}. \tag{33}$$

## A.4 ABLATION STUDIES

Here, we examine the effect of different design choices and hyper-parameters with different ablation studies. In particular, we focus on the 2D Poisson problem in Figure 1 of the main paper.

**Surface point sampling scheme ablations:** Figure 8 compares the deterministic vs. i.i.d. sampling schemes and the effect of various mini-batch sizes (i.e., the number of volumes sampled in each epoch). The results suggest that the deterministic sampling scheme can train successfully with large $N$, however, it may not remedy the biased solution problem with the standard training at small $N$ values. Furthermore, the results indicate that the number of volumes in each epoch has minimal to no effect on the standard training method, which indicates that such a parallelization is not the bottleneck for the standard training method. To make this clear, we showed the training curves for the standard method, and they indicate similar trends and performance across a large range (1-400) of batch-size values for the standard method.

On the other hand, the performance of the delayed target method improves upon using a larger batch size, which possibly indicates that this problem has a high objective estimation variance. The ability to trade large $N$ (i.e., high-quality data) with a larger quantity is an advantage of the delayed target method relative to standard trainings.

**Function approximation ablations:** Figure 9 compares the effect of the neural architecture parameters on the performance of the standard training vs. the delayed target method. All results

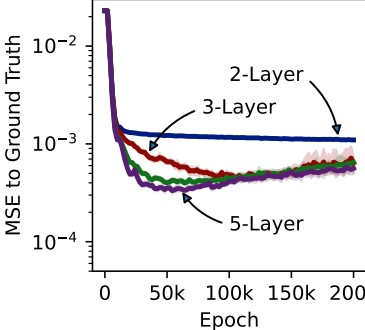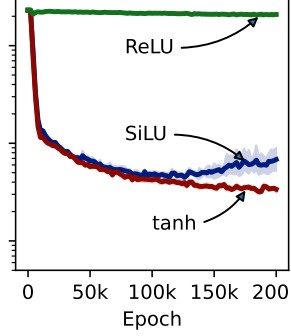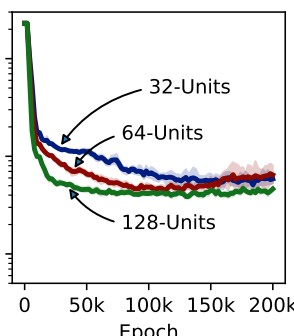

Figure 10: A closer look at the training curves for the delayed target method with different neural network hyper-parameters on the 2D Poisson problem in Figure 1 of the main paper. *The left, middle, and right plots* show the training curves for various neural depths, activations, and widths, respectively.

are demonstrated on the 2D Poisson problem in Figure 1 of the main paper. The standard training exhibits a steady performance across all neural network depths, widths, and activation functions. However, the performance of the delayed target method seems to be enhancing with deeper and wider networks. The effect of the neural activation functions is more pronounced than the depth and width of the network. In particular, the ReLU activation performs substantially worse than the tanh or SiLU activations, and the tanh activation seems to yield the best results.

To shed some further light on the training behavior of the delayed target method, we show the training curves for different neural hyper-parameters in Figure 10. The results indicate that the ReLU activation prevents the delayed target method from improving during the entire training. On the other hand, the SiLU activation yields better initial improvements, but struggles to maintain this trend throughout the training. Based on this, we speculate that some activation functions (e.g., ReLU) may induce poor local optima in the optimization landscape of Algorithm 1, which may be difficult to run away from.

Our neural depth analysis in Figure 10 indicates that deeper networks can yield quicker improvements to performance. However, such improvements are difficult to maintain stably over the entire course of training. In particular, the 2-layer training yields worse performance than the deeper networks, but maintains a monotonic improvement over the course of the training unlike the other methods. Of course, such behavior may be closely tied together with the activation function used for function approximation, as we discussed earlier.

We also show the effect of network width on the performance of the delayed target method in Figure 10. Wider networks are more likely to provide better initial improvements. However, as the networks are trained for longer, such a difference in performance shrinks, and narrower networks tend to show similar final performances as the wider networks.

All in all, our results indicate that the choice of the neural function approximation class, particularly with varying activation and depths, can have a notable impact on the performance of the delayed target method. We speculate that this is due to the incomplete gradients used during the optimization process of the delayed target method. The effect of incomplete function approximation on bootstrapping methods has been studied frequently, both in theory and practice, in other contexts such as the Fitted Q-Iteration (FQI) and Q-Learning methods within reinforcement learning. This is in contrast to the standard training methods, which seem quite robust to function approximation artifacts at the expense of solving an excess-variance diluted optimization problem.

**Integration volume sampling ablations:**  For our integration volumes, we randomly sampled balls of varying radii and centers. The distribution of the sampled radii and centers could impact the performance of different methods. Figure 11 studies such effects on both the standard and the delayed target methods in the 2D Poisson problem of Figure 1 in the main paper. In short, we find that the

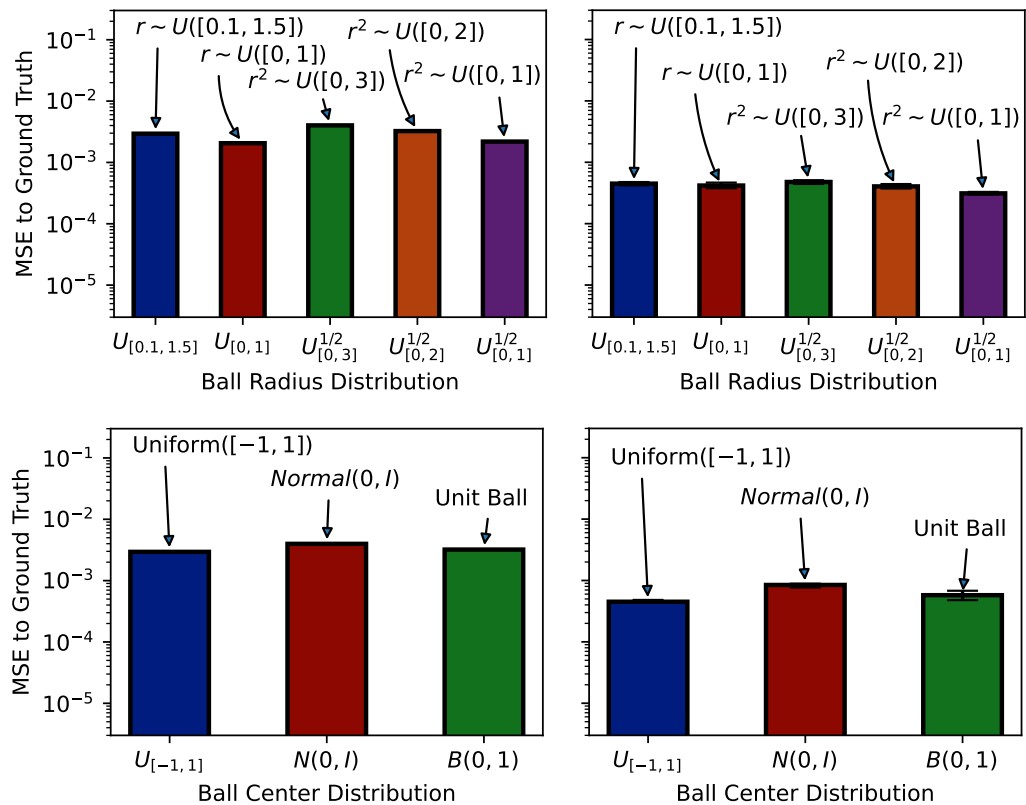

Figure 11: Ablating the distribution of ball centers and radii for both of the standard and the delayed target methods, both with $N = 1$. *The left column plots* correspond to the standard method, and the *the right column plots* correspond to the delayed target method. *The top row plots* show the effect of the ball radius distribution, and *the bottom row* shows the effect of the ball center distribution. In the top row, $r \sim U([a,b])$ means the ball radius was sampled from a uniform distribution over $[a,b]$. Also, $r^2 \sim U([a,b])$ means that the radius was the square root of a uniform random variable between $a$ and $b$. The ball centers were randomly picked either (1) uniformly over a square with the $[-1,-1], [-1,1], [1,1], [1,-1]$ vertices, (2) normally, or (3) uniformly within the unit ball.

delayed target method is robust to such sampling variations; an ideal method should find the same optimal solution with little regard to the integration volume distribution. On the other hand, our results indicate that the standard training performance tends to be sensitive to the integration volume distributions. This may be due to the fact that the standard trainings need to minimize two loss terms; the optimal balance between the desired loss function $\mathcal{L}_\theta(x)$ and the excess variance $\mathbb{V}_{P(x'|x)}[g_\theta(x')]$ in Equation 15 of the main paper may be sensitive to the distribution of $x$ itself.

**Poisson charge placement ablations:** The charge locations in the 2D Poisson problem may impact the results for our methods. For this, we compare the standard and the delayed target method over a wide variety of charge distributions. Figure 12 summarizes these results. Here, the three fixed charge locations shown in Figure 1 of the main paper are shown as a baseline. We also show various problems where the charge locations were picked uniformly or normally in an i.i.d. manner. The performance trends seem to be quite consistent for each method, and the fixed charge locations seem to represent a wide range of such problems and datasets well.

**Robustness with respect to the initial conditions:** Our method is extremely robust to the neural network initializations as shown by the small confidence intervals in our results. In addition, physics-informed networks can readily handle different PDE initial conditions. In fact, the delayed target method is less sensitive to the weight placed on the initial condition loss term, since it can effectively

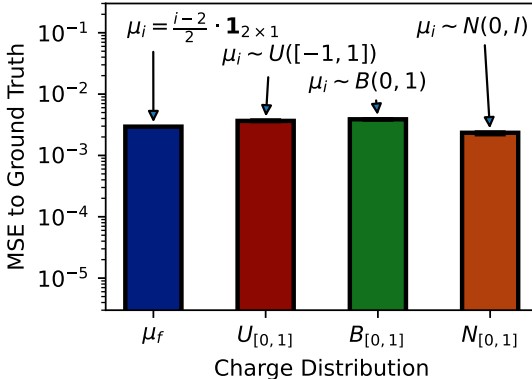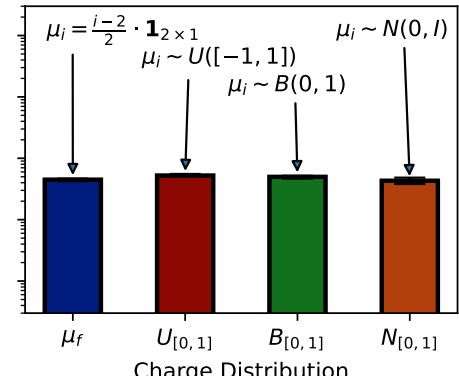

Figure 12: Ablating the location distribution of the three charges on the 2D Poisson problem in Figure 1 of the main paper. *The left plot* shows the results for the standard training method, while the *the right plot* corresponds to the delayed target method, all with $N = 1$. The blue bar represents fixing the charge locations at the $[0, 0]$, $[-0.5, -0.5]$, and $[0.5, 0.5]$ coordinates. We also show the results for picking the charge locations in an i.i.d. manner (1) uniformly between $[-1, -1]$ and $[1, 1]$ (denoted as $U([-1, 1])$), (2) uniformly over the unit ball (denoted as $B(0, 1)$), and (3) normally (denoted as $N(0, 1)$).

eliminate the excess variance term. This is in contrast to the standard training method where the initial condition enforcement may be negatively influenced by the excess variance term.

**Delayed target ablations:** Three main hyper-parameters are involved in the definition of the delayed target method: (1) the target smoothing $\tau$, (2) the target regularization weight $\lambda$, and (3) the target weight $M$ described in Equation 23. Figure 13 studies the effect of each of these hyper-parameters on the performance of the delayed target method in the 2D Poisson problem in Figure 1 of the main paper.

Our results indicate that choosing a proper target smoothing can improve the performance of the delayed target method. In particular, neither a significantly small nor a substantially large $\tau$ can yield an optimal training. Small $\tau$ values cause the training target to evolve rapidly. This may accelerate the training initially, but it can negatively impact the final performance of the method as we show in Figure 13. On the other hand, too large values of $\tau$ can cause the target network to lag behind the main solution, thus bottlenecking the training. The optimal $\tau$ in this problem defines a smoothing window of size $1/(1 - \gamma) = 1000$, which seems relatively appropriate for a training duration of $200k$ epochs.

Next, we studied the effect of target regularization weight $\lambda$ in Algorithm 1. A small target weight causes this method to diverge in this particular problem, as we've shown in Figure 2 of the main paper. On the other hand, a regularization weight too large can slow down the training, as the main model remains too constrained to the target model during training.

We also show the effect of various target weight $M$ values in this problem. Ideally, $M \to \infty$ to make our approximations more accurate. A small target weight can effectively cause the method to seek biased solutions. On the other hand, setting $M$ too large may be impractical and instead cause the loss estimator's variance to explode as discussed in Equation 23 of the main paper. For this, $M$ must be set in conjunction with the $\lambda$ hyper-parameter in such challenging problems.

**Illustration of failure modes:** The delayed target method is more temperamental than the standard training; the set of delayed-target hyper-parameters, such as $\lambda$ and $\gamma$, can have a significant impact on the solution quality. With poor hyper-parameters, the delayed target may poorly track the main solution, and the method can certainly diverge under an inappropriate set of hyper-parameters as we show in Figure 2. Figure 13 of the supplementary material also details the impact of the hyper-parameters related to the delayed target method. Furthermore, Equation 14 indicates that the excess-variance problem can be less severe when the underlying true solution is smooth in $g$

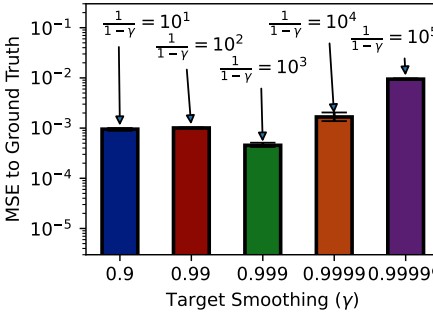 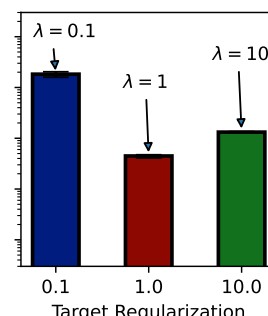 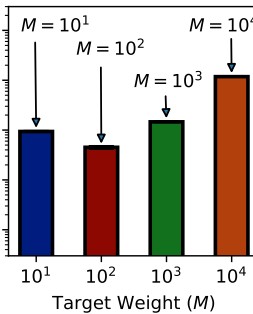

Figure 13: Ablating the effect of the delayed target method parameters on the 2D Poisson problem in Figure 1 of the main paper. *The left plot* shows the effect of the target smoothing parameter $\tau$ in Algorithm 1. *The middle plot* shows the effect of the target regularization parameter $\lambda$ in Algorithm 1. *The right plot* shows the effect of the target weight factor $M$ in Equation 23.

(i.e., when the optimal solution $\theta^*$ has a small $\mathbb{V}_{P(x'|x)}[g_\theta(x')]$ variance). Therefore, when the $g_{\theta^*}$ landscape is nearly flat, we expect the standard training to perform as well as the proposed method.

**Studying the effect of sample sizes:**  Figure A.5 shows the effect of increasing the sample size $N$ on the standard and the delayed target methods. Both methods have improved solution quality with increased $N$, where a 5-fold increase in sample size seems to be more effective for the delayed target method. This confirms that the delayed target method can indeed generate higher-quality solutions at the expense of larger sample sizes.

## A.5   Training Details

**The Poisson Problem**  To solve this system in the integrated form, the standard way of training PINNs is to fit a neural model with the following loss:

$$\hat{\mathcal{L}} = \mathbb{E}\left[\left(A_d^r \cdot \frac{1}{N}\sum_{i=1}^{N} E_\theta(x^{(i)}) \cdot \hat{n}(x_i) - y_\Omega\right)^2\right], \tag{34}$$

where $A_d^r := \int_{\partial\Omega_r} 1 \, \mathrm{d}S$ is the surface area of a $d$-dimensional ball with the radius $r$, and the label is $y_\Omega = \iint_\Omega \nabla \cdot E \mathrm{d}V$.

The $x_i$ samples follow the $\text{Unif}(\partial\Omega_r)$ distribution. The sampling intensity for volume $\Omega$ defines the weight of the test constraints.

In this work, we consider a Poisson problem with $d = 2$ dimensions and Dirac-delta charges. We place three unit charges at $[0, 0]$, $[-0.5, -0.5]$, and $[0.5, 0.5]$ coordinates. For this setup, computing $y_\Omega$ is as simple as summing the charges residing within the volume $\Omega$. The integration volumes are defined as random spheres. The center coordinates and the radius of the spheres are sampled uniformly in the $[-1, 1]$ and $[0.1, 1.5]$ intervals, respectively. We train all models for 200,000 epochs, where each epoch samples 1000 points in total. We also study higher-dimensional problems with $d \in [2, 10]$ with a single charge at the origin.

**Random effect matching**  Random effects (random number generators seed; batch ordering; parameter initialization; and so on) complicate the study by creating variance in the measured statistics. We use a matching procedure (so that the baseline and the proposed models share the same values of all random effects) to control this variance. As long as one does not search for random effects that yield a desired outcome (we did not), this yields an unbiased estimate of the improvement. Each experiment is repeated 100 times to obtain confidence intervals. Note that (1) confidence intervals are small; and (2) experiments over many settings yield consistent results.

**Poisson problem with singular charges**  In this section, we examine solving a singular Poisson problem. In particular, we focus on learning the potential for Dirac-delta charges. Of course, finding

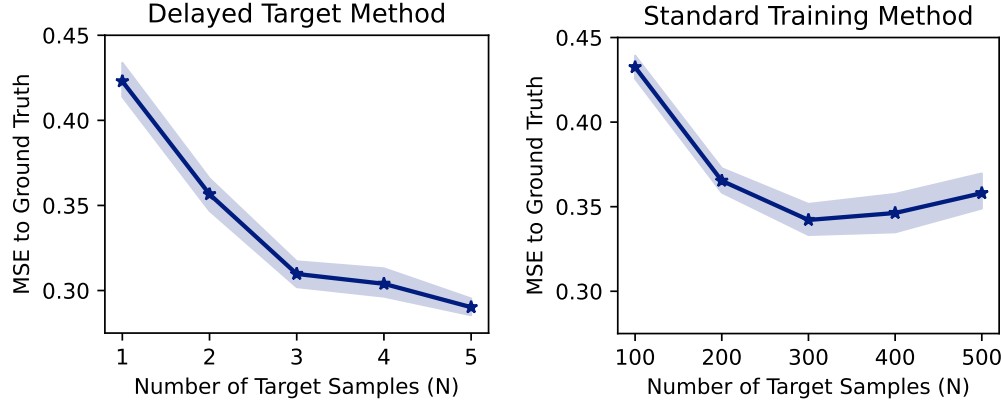

Figure 14: The effect of increasing the number of target samples on the solution quality. Here, the delayed target and standard methods were run for the 10-dimensional Poisson problem of Figure 4 in the main paper while increasing their target sample size $N$ by up to 5-fold. The left plot shows the delayed target method with 1 to 5 target samples and the right plot shows the standard training method with 100 to 500 target samples.

a strong solution by enforcing the Poisson equation directly may be impractical due to its sparsity and singularity. Therefore, we use the divergence theorem as a proxy for learning. This solution relies on estimating an integral over the volume surface, which may require many samples in high-dimensional spaces. Using few samples inside the MSE loss can induce an excess variance that would bias the solution towards overly smooth functions. Our first experiment in Figures 1, 2, and 3 define a 2-D Poisson problem with three unit Dirac-delta charges at $[0, 0]$, $[-0.5, -0.5]$, and $[0.5, 0.5]$. We also study higher-dimensional problems with a single charge at the origin in Figure 4.

**The Maxwell problem with a rectangular current circuit**  Our second example looks at finding the magnetic potentials and fields in a closed circuit with a constant current. This defines a singular current density profile $J$. We consider a rectangular closed circuit in the 3-D space with the $[\frac{1}{\sqrt{3}}, \frac{-1}{\sqrt{3}}, \frac{-1}{\sqrt{3}}], [\frac{1}{\sqrt{3}}, \frac{1}{\sqrt{3}}, \frac{1}{\sqrt{3}}], [\frac{-1}{\sqrt{3}}, \frac{1}{\sqrt{3}}, \frac{1}{\sqrt{3}}]$, and $[\frac{-1}{\sqrt{3}}, \frac{-1}{\sqrt{3}}, \frac{-1}{\sqrt{3}}]$ vertices. The training volumes were defined as random circles, where the center coordinates and the surface normals were sampled from the unit ball, and the squared radii were sampled uniformly in the $[0.0, 1.0]$ interval.

**Smoluchowski coagulation problem**  To simulate particle evolution dynamics, we consider a Smoluchowski coagulation problem where particles evolve from an initial density. We considered the particle sizes $x, x'$ to be in the $[0, 1]$ unit interval, and the simulation time to be in the $[0, 1]$ unit interval as well. We designed the coagulation kernel $K(x, x')$ to induce non-trivial solutions in our unit solution intervals. Specifically, we defined $K(x, x') = 1.23 \times (\min(1.14, \sqrt{x} + \sqrt{x'}))^3$. To find a reference solution, we performed Euler integration using exact time derivatives on a large grid size. The grid time derivatives were computed by evaluating the full summations in the Smoluchowski coagulation equation.

**Training hyper-parameters**  We employ a 3-layer perceptron as our deep neural network, using 64 hidden neural units in each layer, and the Tanh activation function. We trained our networks using the Adam (Kingma and Ba, 2014) variant of the stochastic gradient descent algorithm under a learning rate of $0.001$. For a fair comparison, we afforded each method 1000 point evaluations for each epoch. Table 3 provides a summary of hyper-parameters.

### A.6    LIMITATIONS

Our work solves the problem of learning from integral losses in physics-informed networks. We mostly considered singular and high-variance problems for benchmarking our methods. However, problems with integral losses can have broader applications in solving systems with incomplete

| Hyper-Parameter | Value | Hyper-Parameter | Value |
|---|---|---|---|
| Randomization Seeds | 100 | Problem Dimensions | 1, 2, and 3 |
| Learning Rate | 0.001 | Initial Condition Weight | 1 |
| Optimizer | Adam | Initial Condition Time | 0 |
| Epoch Function Evaluations | 1000 | Initial Condition Points | Uniform([0,1]) |
| Training Epochs | 200000 | Time Sampling Distribution | Uniform([0,1]) |
| Network Depth | 3 | Particle Size Distribution | Uniform([0,1]) |
| Network Width | 64 | Ground Truth Integrator | Euler |
| Network Activation | SiLU | Ground Truth Grid Size | 10000 |

| Hyper-Parameter | Value | Hyper-Parameter | Value |
|---|---|---|---|
| Problem Dimension | 2 | Problem Dimension | 3 |
| Number of Poisson Charges | 3 | Number of Wire Segments | 4 |
| Integration Volumes | Balls | Integration Volumes | 2D Disks |
| Volume Center Distribution | Uniform([-1,1]) | Volume Center Distribution | Unit Ball |
| Volume Radius Distribution | Uniform([0.1,1.5]) | Volume Area Distribution | Uniform([0,1]) |

Table 3: A summary of the general and specific hyper-parameters in each problem. *The top left table* represents the common settings used in all experiments. *The top right, bottom left, and bottom right tables* correspond to the Smoluchowski, Poisson, and Maxwell problems, respectively.

observations and limited dataset sizes. This was beyond the scope of our work. In fact, such applications may extend beyond the area of scientific learning and cover diverse applications within machine learning. We rigorously studied the utility of three methods for solving such systems. However, more algorithmic advances may be necessary to make the proposed methods robust and adaptive to the choice of algorithmic and problem-defining hyper-parameters. The delayed target method was shown to be capable of solving challenging problems through its approximate dynamic programming nature. However, we did not provide a systematic approach for identifying its bottlenecks in case of a failed training. Understanding the pathology of the studied methods is certainly a worthwhile future endeavor.

## A.7 BROADER IMPACT

This work provides foundational theoretical results and builds upon methods for training neural PDE solvers within the area of scientific learning. Scientific learning methods and neural PDE solvers can provide valuable models for a solving range of challenging applications in additive manufacturing (Zhu et al., 2021; Niaki et al., 2021; Henkes et al., 2022), robotics (Sun et al., 2022), high-speed flows (Mao et al., 2020), weather-forecasting (Mammedov et al., 2021), finance systems (Bai et al., 2022) chemistry (Ji et al., 2021), computational biology (Lagergren et al., 2020), and heat transfer and thermodynamics (Cai et al., 2021).

Although many implications could result from the application of scientific learning, in this work we focused especially on settings where precision, singular inputs, and compatibility with partial observations are required for solving the PDEs. Our work particularly investigated methods for learning PDEs with integral forms, and provided effective solutions for solving them. Such improvements could help democratize the usage of physics-informed networks in applications where independent

observations are difficult or expensive to obtain, and the inter-sample relationships and constraints may contain the majority of the training information. Such problems may be challenging and the trained models are usually less precise than the traditional solvers. These errors can propagate to any downstream analysis and decision-making processes and result in significant issues. Other negative consequences of this work could include weak interpretability of the trained models, increased costs for re-training the models given varying inputs, difficulty in estimating the performance of such trained models, and the existence of unforeseen artifacts in the trained models (Wang et al., 2021).

To mitigate the risks, we encourage further research to develop methods to provide guarantees and definitive answers about model behaviors. In other words, a general framework for making guaranteed statements about the behavior of the trained models is missing. Furthermore, more efficient methods for training such models on a large variety of inputs should be prioritized for research. Also, a better understanding of the pathology of neural solvers is of paramount concern to use these models safely and effectively.

