# OpenReview forum: "Learning from Integral Losses in Physics Informed Neural Networks"
_ICLR.cc/2024/Conference — Submitted to ICLR 2024_

### Official Review · Reviewer_LK81 · 2023-10-23

**Soundness:** 2 fair
**Presentation:** 1 poor
**Contribution:** 2 fair
**Rating:** 5
**Confidence:** 3

**Summary:**

This work studies problem of effective stochastic estimation of PDE residuals that occurs during the training of "physics informed neural networks".
The authors prescribe a few techniques that aim to reduce the variance of the underlying parameter gradient. The authors validate their approach on several tasks: Poisson problem, Maxwell problem and a Smoluchowski coagulation problem.

**Strengths:**

- the work studies an important problem of variance reduction in estimation of stochastic dynamical systems
- the paper prescribes a few heuristics that seem to tackle the problem

**Weaknesses:**

- main points of the paper is quite hard to follow or misleading (see Questions)
- the empirical validation is lacking and does not provide a fair comparison with an abundance of the variance reduction techniques used, for instance, in reinforcement learning
- the scale of the problems in the empirical validation is lacking

**Questions:**

- The authors write "The methods we investigate are based around the
idea of reducing the bias and the variance in the parameter gradients": could the authors formally elaborate why so? given the methods formulation it is not clear at all from where specifically "gradient reduction" component comes from.

- Could authors elaborate why the bias-variance like decomposition (that is a common knowledge at this point) occupies almost all of the page 4? I do not see it being a contribution in any meaningful way.

- Could the authors elaborate on the "deterministic sampling strategy": why are you allowed to choose $p(x'|x)$? As it is written, one is most likely to infer that $p(x'|x)$ is defined by the problem itself.

- Regarding the "the double-sampling trick", the authors write "the uncorrelation between $g_\theta(x_1')$ and $g_\theta(x_1')$
will remove the induced bias on average". How come? Could the authors formally show this in the response?

- For "the delayed target method": could the authors elaborate how come (19) holds, in particular, the details regarding the "cross"-term. Also, why does (20) follow? Modulo already a lot of technical assumptions the authors non-rigorously omit, one might easily come up scenario when the set of stationaries of one objective is strictly included in the set of stationary points of the second objective. This, in turn, means that sets of minimizers do not necessarily coincide. Why such scenario does not happen here?

- The authors write "theoretically showed that naive approximations of the parametrized integrals lead to biased loss functions due to
the induced excess variance term in the optimization objective" - this is a bit of a stretch. I have not encountered any formal statement in this paper. In the absence of it, such justification is a mere "heuristics". Could authors point out the rigorous result they are referring to?

---

### Official Review · Reviewer_izgt · 2023-10-31

**Soundness:** 2 fair
**Presentation:** 2 fair
**Contribution:** 3 good
**Rating:** 5
**Confidence:** 4

**Summary:**

In this work, authors proposed a way to find a solution for the problem of training PINNs under partial integro-differential equations. These equations need a tremendous number of evaluations obtained by neural networks to put up a single residual for training. As a result, proper evaluation may be unattainable and replacing integrals with unbiased estimates leads to biased loss functions and solutions. To overcome these restrictions, authors proposed an alternative approach which can be more effective than reducing the variance of the integral estimations. Also, they considered three potential approaches to do this; the deterministic sampling approach, the double-sampling trick, and the delayed target method. Authors also looked at three classes of PDEs to check proposed approach.

**Strengths:**

The authors proposed a novel approach. This work can be a valuable contribution to the field of deep learning-based partial integro-differential equations solvers.

**Weaknesses:**

The composition of the pictures is lacking. It takes a considerable amount of time to grasp the meaning behind the depiction. There is no table of the major results that could have helped with understanding. It is unclear from the work whether this technique can be employed for more applied tasks. The applications are severely limited. The influence of different methods on other models for partial integro-differential equations was not compared, except for multi-layer perception.

**Questions:**

It is necessary to revise the structure of the article regarding the order of introduction of terms and pictures. It should also be reviewed the way the results are presented. Now it is difficult to understand the idea and the results.

---

### Official Review · Reviewer_jJr8 · 2023-10-31

**Soundness:** 4 excellent
**Presentation:** 3 good
**Contribution:** 3 good
**Rating:** 5
**Confidence:** 4

**Summary:**

The paper addresses challenges in training physics-based neural networks when dealing with partial integro-differential equations. Evaluating integral with naive approximation methods leads to bias loss and solutions. As a remedy, the paper proposes three potential solution approaches: the deterministic sampling approach, the double- sampling trick, and the delayed target method. A set of experiments on three benchmark problems confirms the proposed methods (in particular, the delayed target approach) produce solutions that are comparable to the standard sampling approach with many samples.

**Strengths:**

- The paper is overall well-written (but there are some confusions due to the structure and typos, which will be elaborated further in the next item, weaknesses).

- The motivation is clear; an extension of applications of PINNs to integro-de problems.

- The experiments are conducted in a manner that ensures the resulting outcomes serve as compelling empirical evidence showing that the proposed method greatly improves sample efficiency over the regular sampling methods.

**Weaknesses:**

- Some of the proposed methods (the three methods) seem to work well with the benchmark problems. At the same time, some of them do not seem to work very well with the benchmark problems and it is a bit hard to find what would be a take-away message. For example, double-trick does seem to work for the Maxwell problem, but struggles with Poisson problems (Figure 2).  Also, the results of them are not provided for the third benchmark problem. The delayed target method seems to work well with the first two example problems, but for the third problem, it’s hard to see if the proposed one does better than the standard one. In particular, by looking at the training loss curve, it does seem that the proposed method produces higher loss (or is it a coloring issue?). Although there is a small paragraph of recommendation and limitation at the end, it only mentions the delayed target method and it is still less convincing due to the reasons above described.

- The authors provided in-depth ablation studies on the Poisson problem, which can be considered as strength. However, it’s less clear if these findings from the one benchmark problem would also generalize to other intro-DE problems. It would be better if the authors could provide some insights or some paragraph describing tips to set up those hyper-parameters.

- Discussion on computational aspects would also be needed. Computational wall-time would be the best, but at least some discussions on flops would be okay.

- Although the paper is well-structured (section-wise), there are some room for improvements for the presentation.

     - It seems that there should be some break after Eq. (9). Right after Eq. (9) is the description of a loss term for all three examples.

     - Figure 2 describes the results of the delayed target method, but the text reads differently in one place: Page 5 in the paragraph of “The deterministic sampling strategy”, “For a demonstration, Figure 2 shows a number of example sets used for applying the divergence theorem to the Poisson problem.”. Also, in Figure 2, is the pointer to Eq. 24 correct?

     - Figure 6: it’s hard to interpret the figure on the right panel.

     - Recommendation and limitations paragraph: Figure 2 seems to show the performance of the model for varying $M$, not about $\gamma$ or $\lambda$. Should $M$ be interpreted as the hyper-parameters of the proposed methods like $\gamma$ or $\lambda$? $M$ does seem to be a problem-specific parameter. Does this suggest that there can be other problem-specific parameters which needs to be tuned when one want to use the algorithm?

     - The placement of Figures would require many back-and-forth, which makes it difficult to focus.

     -  Conclusion: there is an incomplete sentence: “The delayed target method.”

     - Table 3, is it SiLU? Or Tanh? I think there is a mismatch in the text and the table entry.

**Questions:**

- There are questions in the Weakness section. Please refer to them.

Additional questions would be:

- Would there be any insight or guidance on how to properly choose the hyper-parameter? Or would it be problem specific?

  - Relatedly, would there be any reason why the authors test values of $N$ only from $\{1,100\}$?

- From the related work, it seems that there is some related work to solve the same problem. How would this method compare to those methods? Variational-PINNs (VPINNs) or hp-VPINNs seem the closest ones. Do the authors have a sense on how these related work perform on the benchmark problems?

---

### Official Review · Reviewer_WnMU · 2023-11-01

**Soundness:** 3 good
**Presentation:** 3 good
**Contribution:** 3 good
**Rating:** 6
**Confidence:** 1

**Summary:**

PINNs are used to learn to solve a particular PDEs under the assumption that we can accurately evaluate PDE residuals for learning. In the situations where the PDEs include integrals or large summations, they are evaluated using naive approximations. The authors show these approximations lead to bias solutions. The authors look at three different approaches 1) deterministic sampling approach, 2) double sampling trick, and 3) delayed target method. The authors show that with delayed target method, they achieve comparable results to accurate estimators while using inaccurate estimators. The authors also introduce mitigation strategies to deal with divergence in optimization using delayed target method.

**Strengths:**

- The motivation and overall paper are easy to follow.
- All of their claims seem to be backed by empirical evidence.
- They also introduce a better implementation of the delayed target method which helps with divergence problems during optimization.

**Weaknesses:**

- The author proposed three techniques to reduce the effect of induced bias in PINN, with the result on given examples indicating that the delayed target method has better performance than the others. Is it a universal conclusion or a conclusion under certain conditions? Could the author further explain the benefits and limitations of each technique in comparison to the others?

**Questions:**

Please check the weaknesses above.

---

### Meta-Review · Area_Chair_u7Se · 2023-12-08

**Metareview:**

The authors present a way to extend PINNs from solving partial differential equations to solving integro-differential equations. They show that a naive Monte Carlo approach does not converge well and propose several approaches to solving this problem, with an approach inspired by target networks from deep RL among the most promising. The paper tackles an important problem, which would greatly expand the scope of PINNs if solved. However, several reviewers had concerns that the presentation of the paper was not very clear and it was difficult to evaluate the quality of the results. In particular, it was not clear that the target method was clearly better on all tasks.

I am surprised that the authors did not respond to address these comments. All of the issues the reviewers raised seems like the sort of thing that could be addressed during the review period. Had the authors responded with an improved version of the manuscript, I likely would have recommended acceptance. However, as is, it seems like the manuscript is not ready to be accepted. I hope that the authors will address the comments raised during review and resubmit to another venue.

**Justification For Why Not Higher Score:**

This was addressed in the metareview.

**Justification For Why Not Lower Score:**

This was addressed in the metareview.

---

### Decision · Program_Chairs · 2024-01-16

Reject